# “Lovesick”: Mental Health and Romantic Relationships among College Students

**DOI:** 10.3390/ijerph20010641

**Published:** 2022-12-30

**Authors:** Lacey J. Ritter, Taylor Hilliard, David Knox

**Affiliations:** 1Department of Psychology, Sociology, & Social Work, Mount Mercy University, Cedar Rapids, IA 52402, USA; 2Department of Sociology, East Carolina University, Greenville, NC 27858, USA

**Keywords:** mental health, college students, college relationships, undergraduate students

## Abstract

This research investigated the interpersonal impact of self-reported mental health diagnoses and/or perceptions on undergraduate students’ current or most recent romantic relationship. Analysis of data from a 43-item online questionnaire completed by 267 undergraduates revealed that 68.3% of women and 52.5% of men reported having either been professionally diagnosed with a mental illness or perceive themselves to be mentally ill based on DSM criteria, with women and white students reporting significantly higher levels. Sociologically speaking, mental illness was found to influence relationship initiation, maintenance, and dissolution in this study. The mental health of the respondents’ potential partners was an important consideration in deciding to form a relationship, particularly for male, white, heterosexuals. When the respondents reported relationship problems, men were more likely to blame such problems on mental health issues than women. Finally, though more respondents reported having broken up with a romantic partner who had mental health issues than had romantic partners break up with them, there were no significant gender, race, or sexual orientation differences in the termination of these romantic relationships. Study findings emphasize the importance of acknowledging and providing mental health resources—particularly interpersonal options—for emerging adults in the college setting.

## 1. Introduction

Relentless mass shootings continue to reflect the reality of mental illness as a serious problem in our society. It is not unusual that individuals struggle with severe depression/anxiety—particularly in the post COVID-19 era—and many seek medical or psychological services for relief. The stigmas associated with mental illness, however, can prevent medical reliance by those who also have symptoms, but no official diagnosis [1]. While society often considers mental illness to be an individual experience, the social impact of mental health on interpersonal relationships, particularly romantic relationships, is the focus of this research.

Mental illness, particularly among college students, has been the subject of considerable research. Barnett et al. [1] noted that 12% to 50% of college students meet the criteria for one or more mental health disorders (e.g., anxiety, depression), and Sherman [2] confirmed that increasing numbers of college students exhibit mental health symptoms. One of the contexts in which students experience mental illness is in their romantic relationships.

In this study, we sought to understand how undergraduate students’ struggle with mental illness impacts their romantic relationships. Specifically, we tested whether either clinically diagnosed or self-perceptions of mental illness were associated with initiating or dissolving a romantic relationship and how respondents felt their mental health was shaped—and changed—by their relationships. Data from a 43-item survey created by the authors was used to analyze the interpersonal impact of mental health on romantic relationships. Understanding the association between mental health and romantic relationships is important in helping college students improve their mental well-being, enhance educational success, and maintain/create positive social relationships. Since emerging adults are expected to form long-lasting romantic connections either during or after college, the influence of mental health on these relationships is an important area of scientific inquiry.

Mental illness is a pervasive social problem, with recent estimates revealing that approximately 19% of U.S. adults experience mental illness, with 4% reporting severe forms [3] and serious psychological distress [4] in the past month. Young adults, in particular, have a higher prevalence of mental illness, with young female adults suffering the highest levels [4]. Over the past decade, the percentage of young Americans experiencing certain types of mental health disorders has increased significantly [5]. Data reveal a 63% increase of symptoms consistent with major depression among young adults ages 18 to 25 in the past decade, with similar increases in levels of serious psychological distress/suicidal thoughts [5].

Considerable literature confirms that mental health and wellness have a dramatic impact on overall health and life outcomes, with evidence showing the inverse association between physical and mental health outcomes [6]. Overall, mental illness—and the related effects on physical health—is a major public health concern, costing billions of dollars in direct health costs and indirect costs. Untreated mental illness increases risky behaviors in sufferers including suicidal ideations and, unfortunately, suicidal deaths—the 2nd leading cause of death among adults ages 25 to 34 [7]. The mass public and school shootings in our society reflect compromised mental health in the shooter.

College students report depression, anxiety, and stress [8], with approximately one in three undergraduate students revealing levels of depression high enough to impede function, as well as nearly one in 10 students indicating that they were high-risk for suicidal attempts in the previous year [9]. Despite these data, the importance—and treatment—of mental health issues, particularly for college-aged young adults, are not widely recognized by policy makers, health-care providers, and the general public [6]. Indeed, research on coping strategies and resilience show promise (for example, see [10,11]), but are often ignored for this age group, considered to be in the prime of their lives, yet also not quite mature enough to be suffering from “real” problems associated with adulthood [12]. The following literature review provides a glimpse into research on the interpersonal mental illness experiences of college students, particularly the impact of perceived and/or diagnosed mental illness on their social and romantic relationships.

### 1.1. Mental Illness Trends in College Students

Within the young adult population, college students experience mental health problems at alarming rates, with recent surveys finding that over half of students reporting overwhelming anxiety, and over 1/3 reporting depression in the past year [13]. Indeed, the literature suggests that a wide range of emotional problems, beyond depression, are significantly associated with lower academic functioning [14]. Additional outcomes include increased physical and sexual health risks, juvenile delinquency, underemployment, substance abuse, unhealthy weight gain, and premature mortality [12,15,16]. Students of color face additional mental illness risks due to perceived discrimination, especially at primarily White institutions (PWIs), tied to race-related stressors and unease during their college experience [17,18].

Among college students of all racial/ethnic backgrounds, there is an especially high prevalence of alcohol use disorders [19,20]. Alcohol use disorders often lead to lifelong mental and general health consequences [19,21]. Since many of these same at-risk young adults do not seek mental health services for mental illness [22], the risk of developing long-term consequences—or relying on informal, untrained sources of support, is high [22].

Some research, however, shows that college students are at a slightly lower risk of mental disorder in comparison to their non-college peers [14]. This is, in part, due to universities across the country conducting various research studies, providing services, and implementing various techniques to improve students’ mental health. Attempting to provide more awareness, many colleges are beginning to introduce mental health information with students as early as orientation sessions and within courses, making students aware of symptoms; emphasizing resources available to them, teaching wellness/well-being practices, and preparing them to support friends and classmates who might be struggling with mental illness [23,24,25]. Indeed, prevention and early intervention (PEI) not only increases the likelihood that students use mental health resources and do so sooner, but also lowers corresponding student dropout rates due to, among other reasons, poor mental health [15,25]. By creating a positive campus mental health culture and reducing the barriers to treatment, there is the potential for an increase in students seeking treatment [13], as well as using their own coping skills and resilience to help bolster their mental health [11].

### 1.2. Mental Health and Social Relationships: The Good, The Bad, and The Isolated

A major buffer for mental illness, social support—the intention of providing help and being there for others within social relationships—is of four types: emotional, instrumental, informational, and appraisal [26]. Prior to college, teen and adolescents get most of their support from family and peers, though the importance of relying on family does not stop when arriving at college—indeed, parents helping their college-enrolled children learn healthy independence while assisting with demands and challenges related to both college and impending adulthood are not without merit [27]. Even if parents are no longer physically present, the perception of support functions to bolster mental health [28,29].

Though college students are benefitting from continued relationships with their parents, they may also be gaining social support from others on campus, whether classmates, roommates, or faculty members [30]. Having individuals to provide advice, talk through problems or struggles, or even simply to “be there” for students is associated with their likelihood to seek and continue treatment [23,31]. These combinations of helpful networks increase the likelihood that students will remain enrolled in college/university [30]; along with their abilities to be resilient and to develop intimate social relationships with others, the focus of this research.

Social support, however, may not always be beneficial. If students’ parents also suffer from mental illness [28]; there is a breakdown of communication or a history of abuse and intergenerational problems in the student’s family [23] or there is a barrier to developing social relationships with others [9], the perceived lack of adequate social support will instead impede students’ ability to reach out—and receive—mental health help. In particular, levels of perceived support from family have been negatively associated with physical health [32]. Similarly, negative perceptions of support from peers and romantic partners are associated with higher levels of loneliness, depression [32], anxiety, insecurity, and low moods [33]. These health problems may compound to produce low self-esteem [34].

Dealing with mental illness is difficult enough, but the label and stigma of mental illness can make perceptions of social support—and the confidence to reach out for help and treatment—even more difficult. Feeling that others have also applied the deviant label of “mentally ill” to them, students may withdraw from other students or sources of support on campus [35], particularly if campus culture and media portrayals teach them that mental illness is being weak or unacceptable. [36]. While this may prevent students from seeking the mental help they need, it may also perpetuate the idea that mental illness should be hidden—reiterating this gap in help-seeking [37].

In addition to being considered deviant, the mentally ill are stigmatized for their illness, leading to higher risk of negative health and social outcomes for students. Indeed, the stigma of mental illness is the fourth most common barrier for seeking mental health treatment for young adults [13]. The failure to seek treatment or the fear of their stigma being revealed and shared with others, or of their illness contaminating the “healthy” partner [13,38], makes the risk factors associated with mental illness in college even more concerning.

### 1.3. Mental Health and Romantic Relationships—Creating a Research Focus

In addition to increased mental health risks associated with the transition to college, romantic relationships are sometimes associated with increased risk of mental illness [39]. Specifically, breaking up or losing a romantic partner increases one’s levels of psychological distress [40], which can compound already-existing mental health concerns, particularly depression and substance use (for not having a love interest at all) [41] when compared to their coupled counterparts [42,43].

By contrast, research on the benefits of romantic relationships has been found to serve as a buffer to negative mental health outcomes [42,44,45], particularly for women’s levels of depressive symptoms [46]. Even intrapersonal affects have been noted, with romantic relationships boosting self-esteem and self-concepts for an individual [43,47]. Whether through monitoring changes in their partner’s behavior, providing general social and emotional support, or advocating for health care or therapy [48,49], the value of relationships on mental health outcomes is impressive.

Young adults transitioning to college with unmet mental health needs impacts their ability to maintain healthy romantic relationships [13]. Though relationships can help improve mental health and may address some of these unmet needs, generally, committed, longer-lasting relationships are the ones most beneficial to mental health outcomes [50,51]. Gendered benefits in mental health outcomes related to relationship status similarly find college-aged women at increased risk of mental illness following a breakup [43]. For college students of any gender, however, the impact of relationship dissolution on mental health tends to add another type of stressor into this already tumultuous time, particularly if the breakup is internalized and exacerbates already-existing mental health problems [11].

### 1.4. Summary and Hypotheses

This study sought to answer whether mental health was associated with relationship creation, maintenance, and dissolution among undergraduate college students. We tested the following hypotheses based on prior research:

**Hypothesis** **1.**
*Social location will influence rates of mental health problems. In particular, minority students (females, racial/ethnic minorities, and LGBT students) will report higher rates of mental health diagnoses than their majority counterparts. Existing literature has confirmed that young adult women face higher mental illness rates than their male counterparts [4], LGBTQ individuals are twice as likely to have a mental disorder compared to heterosexual individuals [52] and students of color are vulnerable to increased mental illness risk because of perceived discrimination [17,18].*


**Hypothesis** **2.**
*Students will report a preference for romantic partners who do not experience mental illness. Existing literature shows that individuals see those with mental illnesses as having lower than average potential as a partner, particularly for a long-term relationship [53]. However, minority students (females, racial/ethnic minorities, and LGBT students) will be more accepting of dating a romantic partner who has mental illness than heterosexual, White, male respondents.*


**Hypothesis** **3.**
*Students will report that their mental health either increased or was not changed by their romantic relationships. This hypothesis is based on past studies of college students in serious romantic relationships who reported increased mental health due to the increased social support provided by partners [42]. Increases in self-esteem and individual well-being have also been associated with romantic involvement [47].*


**Hypothesis** **4.**
*Mental illness will not be a common cause of relationship dissolution for most college students. Existing literature has empathized that—as mentioned above—relationships improve the mental health of partners. Thus, respondents would be less likely than their single counterparts to suffer from mental illness or break up because of their context [42,43].*


## 2. Materials and Methods

### 2.1. Participants

The authors developed a 43-item survey approved by the Institutional Review Board at a public university in the southeastern United States and posted it online in the fall of 2019. Students in the third author’s Courtship and Marriage, as well as the Marriage and Family course, were emailed the link and asked to complete the survey. Sociology colleagues/faculty also sent the survey link to the students in their introductory courses. Questions regarding gender, sexual orientation, race/ethnicity, religiosity, and life-satisfaction preceded questions about the respondents’ mental health status and roman tic relationships. A total of 277 students initially completed the survey, providing a response rate of approximately 86%. After removing respondents from the sample who both reported that they were not undergraduate students at the time of the survey or who did not report information on the main dependent variable—mental health diagnoses—the survey sample consisted of 267 students.

### 2.2. Measures

#### 2.2.1. Relationship Formation

Relationship formation was measured by asking respondents to agree or disagree with the following statement: “I prefer to have a relationship in which my partner does not have mental health issues” using a 5-point Likert scale in which 1 = Strongly Disagree and 5 = Strongly Agree. Higher scores represented agreement that mentally healthy partners were preferred.

#### 2.2.2. Relationship Maintenance

The association between relationship maintenance and mental health was measured by three variables. First, using the same Likert scale measure, respondents were asked whether they agreed or disagreed with the following five initial statements: (1) Being in a romantic relationship has improved my mental health; (2) I sometimes blame problems that I have in my romantic relationships on my partner’s mental health; (3) I sometimes blame problems that I have in my romantic relationships on my own mental health; (4) If problems occur in my romantic relationship, my partner sometimes blames their own mental health issues; and (5) If problems occur in my romantic relationship, my partner sometimes blames my mental health issues.

*Mental health improvement* focused on the first statement, ranging from 1 to 5, with 5 being strong agreement that the respondent’s mental health improved with their romantic relationship. The final four questions were combined into two variables: *blaming the partner’s illness* (based on statements 2 and 4; Chronbach’s alpha (α) = 0.61) and *blaming the respondent’s illness* (based on statements 3 and 5; α = 0.60), with possible scores ranging from 2–10. These alpha scores are poor at best; however, initial tests using each question separately (5 measures instead of 3) found no statistically significant differences in outcomes, so blame variables were combined to reduce models for the manuscript. Initial tests are available by request.

#### 2.2.3. Relationship Dissolution

Analysis related to ending a romantic relationship included three variables in regard to ending a relationship.” *Partner’s dissolution*” asked respondents whether they agreed or disagreed with the following statement: A partner has broken up with me because of my mental health issues. “*Respondent’s dissolution*” asked a similar question, focusing on the respondent’s ending a relationship due to their partner’s mental health issues. Both measures ranged from 1–5, with higher scores indicating stronger levels of agreement. Finally, respondents were asked how many of their romantic relationships had ended due to mental health issues on behalf of one—or both—partners. For each of the two separate questions, respondents reported whether none, one, two, three, or four or more relationships had primarily ended due to their or their partner’s mental health issues. The two variables were combined to show a total count of the number of relationships ended due to mental health, ranging from 0 to 8, with higher values representing more relationships having ended due to mental illness.

#### 2.2.4. Control Variables

Of the students sampled, *gender* was a dichotomous measure in which (83.9%) of the respondents were female; 15.0% male. *Race*/*ethnicity* was measured by having respondents report their racial/ethnic background. Responses included: 66.67% white, 16.48% black, 6.37% Hispanic, 1.87% Native American, 3% Asian, and 5.61% multiracial (some respondents self-identified in more than one category). For analysis, race was dichotomized into white (66.67%) and racial/ethnic minority (33.33%) students. *Sexual orientation* was assessed by asking respondents to report their self-identified sexual orientation based on the following options: heterosexual (87.59%), gay/lesbian (3.01%), bisexual (6.39%), and asexual/queer/other (3.01%). Sexual orientation was dichotomized into heterosexual vs. sexual minorities for analysis. Students also reported their *grade level* based on the amount of credit hours they had completed. 68.54% of the respondents were freshmen or sophomores, and 31.46% of the respondents were juniors or seniors. The average *age* of respondents was 19.34.

### 2.3. Procedure

T-tests were used for the initial analyses of mental illness diagnoses followed by Spearman’s correlation coefficients (and Spearman’s rho) to test for differential diagnoses rates and strength of association by gender, race, sexual orientation, and education level. Both individual diagnosis rates and overall index scores were tested by social location. Findings are described below.

Secondary analyses looked at relationship formation, maintenance, and dissolution using *t*-tests and Spearman’s correlation coefficients (and Spearman’s rho) in three models. Model 1 tested relationship initiation, comparing rates across gender, race, and sexual orientation. Model 2 tested relationship maintenance and Model 3 tested relationship dissolution, comparing rates across gender, race, and sexual orientation. Since the initial analyses found no significant differences in mental health diagnoses by grade level, secondary analyses were not conducted on this variable.

## 3. Results

### 3.1. Initial Analyses—Mental Health Diagnoses and Rates by Social Location

Over half (65.9%) of the sample reported that they had one or more mental health symptoms, ranging from the most frequently reported diagnoses of anxiety (47.57%) and depression (34.83%) to schizophrenia, Autism, and bipolar disorders (0.37% for each). The average number of reported mental disorders was 1.27 conditions (SD = 1.27). Being female, a sexual minority, and an upperclassman were associated with higher frequencies of reporting mental health problems.

Females were significantly more likely than males to report mental health diagnoses (1.34 compared to 0.85, respectively) (*p* < 0.05), providing partial support for Hypothesis 1. Though not significant (*p* = 0.3177)., heterosexuals reported an average of 1.25 mental health conditions in contrast to sexual minority respondents (gay, lesbian, bisexual, asexual, and other) who reported an average of 1.48 mental health conditions (*p* = 0.3177). Overall, racial/ethnic minorities reported significantly fewer mental health diagnoses than white respondents (1.056 compared to 1.382 diagnoses, respectively) (*p* < 0.05), contrary to existing literature and Hypothesis 1. Juniors and seniors reported slightly more (1.31) mental health diagnoses than did their underclass (freshman and sophomore) counterparts (1.26 diagnoses), though the difference was not significant (*p* = 0.75). Table 1 shows the percentages of each specific mental illness reported by the total sample and by social location.

There were significant differences across three of the four measures of social location. Women were significantly more likely than men to have been diagnosed with anxiety (*p* < 0.001) and eating disorders (*p* < 0.05) and had significantly more diagnoses overall than men (1.339 and 0.85, respectively) (*p* < 0.05). Men, by comparison, were significantly more likely to have been diagnosed with ADHD and schizophrenia than women (*p* < 0.05). Dividing respondents across sexual orientation, sexual minority (LGB) students were more likely to have been diagnosed with autism (0.030) and extreme phobias (0.061) than heterosexual students (*p* < 0.05), though overall indices for mental illness diagnoses were not significantly different by sexual orientation. White students reported significantly more mental health diagnoses than racial/ethnic minority students (1.382 and 1.056, respectively) (*p* < 0.05). Specifically, white students reported significantly more ADHD (0.157, *p* < 0.001), anxiety (0.533, *p* < 0.001), and eating disorder (0.107, *p* < 0.05) diagnoses, while racial/ethnic minority students reported significantly higher levels of “other” mental disorders (0.090, *p* < 0.05). There were no statistically significant differences in mental health diagnoses by student grade level.

### 3.2. Secondary Analyses—Mental Health and Relationship Formation

In regard to beginning a new relationship, the mental health of a potential romantic partner can impact one’s interest/decision in becoming involved with the person. Students were asked whether they agreed or disagreed with the following statement: “I prefer to have a relationship in which my partner does not have mental health problems.” While approximately half of students reported that they neither agreed nor disagreed with the statement, the second-largest group of respondents agreed with the statement (18%), 11% strongly agreed.

Students, on average, reported median, neutral levels of agreement with the role of mental health in deciding to become involved with a romantic partner. Breaking down the results by social location demonstrated for which students a partner’s mental health mattered. Table 2 reveals the averages and Spearman’s correlations for each measure of relationship influence by gender, race/ethnicity, and sexual orientation.

In deciding whether the mental health of a potential romantic partner influenced the formation of a relationship, heterosexual students were significantly more likely (3.132) to report a preference for a partner without a mental illness diagnosis than sexual minority respondents (2.655) (*p* < 0.05). Women were also slightly less likely (3.036) than men to agree that they preferred having a romantic partner without mental health problems (3.314), though the finding was not significant. Differences by race/ethnicity were also not significant, though white respondents were more likely to agree that mentally healthy partners were preferred.

### 3.3. Secondary Analyses—Mental Health and Maintaining Relationships

Once involved in a romantic relationship, mental illness remained a significant issue. In general, respondents in romantic relationships reported fairly low average levels (3.95) of a partners’ mental illness issues causing relationship problems. When problems did exist, respondents and their partners were more likely to blame themselves, with an average score of 4.83 out of 10. However, blaming one’s partners’ mental illness for relationship problems was significantly more likely for males (4.500) than females (3.842) (*p* < 0.05). Differences by race and sexual orientation in blaming a partners’ mental illness on relationship problems were not significant. Students were also asked to agree or disagree as to whether being in a romantic relationship had improved their mental health, with higher scores representing stronger agreement. Overall, students reported that being in a romantic relationship had either no influence (32%) or had a positive impact on their mental health (33%). Approximately 12% of students, however, either strongly disagreed (3%) or disagreed (9%) that their mental health had improved since their involvement in a romantic relationship. White students were significantly more likely to agree that their mental health had improved since being in a romantic relationship (3.774) than racial/ethnic minority students (3.327) (*p* < 0.01). Differences in one’s mental health improvement in reference to a romantic relationship were not significant by gender or sexual orientation.

### 3.4. Secondary Analyses—Mental Health and Relationship Dissolution

To better understand relationship dissolutions, respondents were asked how many of their romantic relationships had ended due to mental health issues on behalf of one—or both—partners. Overall, most students had not ended a romantic relationship due to a partner’s mental illness (73%). For those who had ended a romantic relationship due to mental health, most (24%) had only ended one or two relationships. Only 6 students (less than 3%) reported 3 or more relationship dissolutions due to mental illness. There were no significant differences in the number of relationships that dissolved due to mental illness by gender, race, or sexual orientation.

## 4. Discussion

Results revealed that, while over half of these 267 undergraduates reported some form of personal mental illness diagnosis, the mental health of their romantic partners played a role in their relationship initiation, maintenance, and dissolution. Particularly for white, heterosexual men, finding relationship partners who do not suffer from mental illness was important, supporting stigma research that even stigma by association is seen negatively [54]. By contrast, women, non-whites, and sexual minority respondents were, on average, less likely to agree that finding a mentally healthy romantic partner was important. Previous researchers have suggested that minority individuals (women, racial/ethnic minorities, LGBT) are more likely to be accepting of other minority individuals due to both contact hypotheses [55] and minority-stress experiences [56].

Regarding the impact of one’s romantic relationship on their mental health, most students agreed that their mental health either remained the same or improved, supporting previous research on the value of social support of partners [42,43]. If there were relationship problems, men were significantly more likely to blame their partner’s mental health issues for relationship conflicts than were women. Social scripts surrounding gender roles [57] and romantic relationships [58] often blame women for problems and difficulties, particularly since men are expected to be tough, strong, and—above all—not “crazy.” Indeed, a comparison of 718 female and 3627 male criminal court-ordered psychiatric evaluations revealed that women were more often evaluated as having a mental disease or defects [59].

Finally, though more respondents reported having broken up with a romantic partner who had mental health problems than had been broken up with, there were no significant gender, race, or sexual orientation differences in dissolution. Involvement in a romantic relationship may also have a positive effect on a person with mental health problems. Previous researchers have noted how mental illness within a relationship can be improved through the intimate social support gained from a loving partner [42,43] and from a partner’s emphasis on health-positive behaviors, including communication, seeking professional help, and improving physical health as a tool for better mental health outcomes [60]. Though the researchers did not ask why the respondents dissolved the relationship due to mental health, the likelihood of having done so emphasizes the importance of further study to discover these narratives.

### 4.1. Limitations

The data should be interpreted cautiously. First, regarding the undergraduate sample, the data were skewed toward females (83.9%) and Whites (66.7%). The sample of 267 undergraduates is small—calling into question the power and usefulness of the sample size. The sample—one of convenience and suffering from self-selection bias given its administration through academic courses—is also not generalizable to the 15 million college and university students throughout the United States [61]; is not generalizable to non-enrolled young adults or other age groups; and is not generalizable to international populations who have different norms and experiences related to mental illness and interpersonal relationships. Future research should incorporate larger samples as well as students from private and religious affiliated universities both inside and outside the United States. International comparisons would shed light on comparative prevalence rates; impacts on interpersonal relationships; and ways of combatting negative impacts of mental illness on social wellness. The occurrence, however, of mental illness—and interpersonal relationships—during emerging adulthood is of global concern, given that this population will be heading into economic sectors and possibly be starting families—impacting other individuals and institutions with their mental health outcomes. Cross-cultural and international comparisons will provide awareness into prevalence—and hopefully provide solutions—that can be used at a global level to reduce stigma and unmet mental health concerns.

Another limitation of the study is that it was cross-sectional. Respondents were asked to report on mental health status and its impact in reference to their current or previous relationship at the time they took the survey. Particularly for students who were reporting on past relationships, their interpretations of mental health’s influence on their romances may be biased or interpreted differently based on relationship outcomes. For students currently in a romantic relationship, they were also reporting at one period in time, making assessments of fluctuations over time and perceptions of long-term effects on relationship stability related to mental health issues impossible. Future studies on mental health impact should follow individuals over time to monitor relationship initiation, maintenance, and dissolution patterns.

Finally, the study relies on a combination of self-perceived mental illness and physician diagnoses. The Diagnostic and Statistical Manual of Mental Disorders—and criteria for diagnoses—have changed throughout its various iterations. Since the study took place in 2019, diagnoses likely came from the DSM-5, published in 2013, and relied on ICD-10, which was created in 1992. Since the majority of senior students would have been born approximately in 1998 at this time, we feel confident that reliance on the DSM-5 would have been used. Physician changes in diagnosis would have been accounted for in student answers—lay reliance on “common knowledge” about mental health—which again will play a significant role in opinions and treatment of self and others—may not match the most updated classifications of symptoms. Future research could rely exclusively on official physician diagnoses or ask students to complete the battery of ICD-10 classifications to determine whether they would qualify for a particular diagnosis at time of survey.

### 4.2. Conclusions

Though many respondents reported various levels of their own mental health, their expectations for their partners seem to favor the mentally healthy, particularly for White, heterosexual students. Though one’s own mental health was found to either not change or to improve within a romantic relationship, students reported ending or having a relationship ended because of mental illness symptoms. Future research should examine these trends with larger samples and at variously affiliated universities. Knowledge of how mental health is associated with romantic relationships in college increases our understanding of the variables involved in initiating, maintaining and dissolving relationships as college students continue their path of emerging adulthood.

Practical and educational implications of the study are two-fold. First, young adults are often considered to fare well on their own, without aid for either mental health or romantic relationships [12]. This is a period in which mistakes and experiments (along with their messes) are expected. As noted above, college students report depression, anxiety, and stress [8], with approximately one in three undergraduate students revealing levels of depression high enough to impede function, as well as nearly one in 10 students indicating that they were high-risk for suicidal attempts in the previous year [9]. Institutions of higher education should incorporate better models for mental wellness into their curriculum to both raise awareness of and normalize mental health struggles in ways that teach students to seek help—and access it when and where it is offered. For example, providing free mental health counseling for students and, in the case of this study, interpersonal counseling options, is one way to combat negative educational and interpersonal outcomes of mental illness [62,63,64].

Second, large-scale implications of mental illness impact more than just college students. As noted above, the financial costs of mental illness are a major public health concern [7]. In addition, the interpersonal and social costs of symptoms and their interpersonal effects—such as suicide or mass shootings—is an additional burden to society and its institutions. Though this study focused only on college students in the United States, mental health implications are not U.S.-specific. Cross-cultural comparisons will broaden understanding of the interpersonal impact of diagnosed or self-perceived mental illness in emerging adult relationships in ways that provide additional insight and possible remedies to this phenomenon.

## Figures and Tables

**Table 1 ijerph-20-00641-t001:** Initial Analyses of Mental Health Conditions in College by Social Location.

		Gender		Sexual Orientation		Race/Ethnicity		Grade Level	
Mental Health Diagnosis	Total Sample (*n* = 267)	Males (*n* = 40)	Females (*n* = 224)	Hetero- Sexuals (*n* = 233)	Sexual Minorities (*n* = 33)	Whites (*n* = 178)	Racial Minorities (*n* = 89)	Fresh. /Soph. (*n* = 183)	Junior /Senior (*n* = 84)
ADHD (0,1)	11.61% *n* = 31	0.225 * (0.067)	0.094 * (0.020)	0.120 (0.021)	0.091 (0.051)	0.157 ** (0.027)	0.034 ** (0.019)	0.137 (0.025)	0.071 (0.028)
Depression (0,1)	34.83% *n* = 93	0.225 (0.067)	0.371 (0.032)	0.348 (0.031)	0.364 (0.085)	0.365 (0.036)	0.315 (0.050)	0.333 (0.035)	0.381 (0.053)
Anxiety (0,1)	47.57% *n* = 127	0.200 *** (0.064)	0.527 *** (0.033)	0.472 (0.033)	0.515 (0.088)	0.533 ** (0.037)	0.360 ** (0.051)	0.443 (0.037)	0.548 (0.055)
Autism (0,1)	0.37% *n* = 1	0 (0.000)	0 (0.000)	0 ** (0.000)	0.030 ** (0.030)	0 (0.000)	0.011 (0.011)	0.005 (0.005)	0 (0.000)
Schizophrenia (0,1)	0.37% *n* = 1	0.025 * (0.025)	0 * (0.000)	0.004 (0.004)	0 (0.000)	0.006 (0.006)	0 (0.000)	0.005 (0.005)	0 (0.000)
Bipolar (0,1)	4.49% *n* = 12	0.075 (0.042)	0.036 (0.012)	0.047 (0.014)	0.030 (0.030)	0.045 (0.016)	0.045 (0.022)	0.049 (0.016)	0.036 (0.020)
Eating Disorders (0,1)	8.24% *n* = 22	0 * (0.000)	0.982 * (0.020)	0.082 (0.018)	0.091 (0.051)	0.107 * (0.023)	0.034 * (0.019)	0.087 (0.021)	0.071 (0.028)
Substance Abuse Disorders (0,1)	0.75% *n* = 2	0 (0.000)	0.004 (0.004)	0.004 (0.004)	0.030 (0.030)	0.006 (0.006)	0.011 (0.011)	0.011 (0.008)	0 (0.000)
Post-Traumatic Stress Disorders (PTSD) (0,1)	4.87% *n* = 13	0.025 (0.025)	0.054 (0.015)	0.043 (0.013)	0.091 (0.051)	0.051 (0.016)	0.045 (0.022)	0.055 (0.017)	0.036 (0.020)
Extreme Phobias (0,1)	1.5% *n* = 4	0 (0.000)	0.018 (0.009)	0.009 * (0.006)	0.061 * (0.042)	0.011 (0.008)	0.022 (0.016)	0.022 (0.011)	0 (0.000)
Obsessive–Compulsive Disorder (OCD) (0,1)	7.49% *n* = 20	0.025 (0.025)	0.085 (0.019)	0.077 (0.018)	0.061 (0.042)	0.067 (0.188)	0.090 (0.030)	0.066 (0.018)	0.095 (0.032)
Other (0,1)	4.87% *n* = 13	0.05 (0.035)	0.491 (0.014)	0.043 (0.013)	0.091 (0.051)	0.028 * (0.012)	0.090 * (0.030)	0.044 (0.015)	0.060 (0.026)
Index Total:	-	0.85 * (1.00)	1.339 * (1.26)	1.249 (0.083)	1.485 (0.227)	1.382 * (0.099)	1.056 * (0.119)	1.257 (0.099)	1.310 (0.121)

Note: For means, standard deviations are in parentheses; * *p* < 0.05; ** *p* < 0.01; *** *p* < 0.001.

**Table 2 ijerph-20-00641-t002:** Spearman’s Correlation Coefficients of Mental Illness Problems and Romantic Relationships ^1^.

	Model 1— Relationship Formation	Model 2— Relationship Maintenance			Model 3— Relationship Dissolution		
	*Mentally* *Healthy* *Partners*	*Mental* *Health Improved*	*Blame Partner’s Illness*	*Blame* *Respondent’s Illness*	*Partner’s Dissolution for Mental Illness*	*Respondent’s Dissolution for Mental Illness*	*Number of* *Relationships Ended*
**Gender**							
Females (0,1)	3.036 (0.076)	3.617 (0.083)	3.842 * (0.146)	4.841 (0.157)	1.898 (0.082)	1.968 (0.094)	0.378 (0.054)
Males (0,1)	3.314 (0.196)	3.821 (0.179)	4.500 * (0.373)	4.667 (0.412)	1.885 (0.224)	1.889 (0.209)	0.429 (0.155)
	*−0.0873*	*−0.0696*	*−0.1279*	*0.0380*	*0.0259*	*0.0133*	*0.0027*
**Race** **/Ethnicity**							
Whites (0,1)	3.101 (0.081)	3.774 ** (0.084)	3.877 (0.159)	4.948 (0.170)	1.948 (0.094)	1.896 (0.099)	0.446 (0.065)
Racial /Ethnic Minori ties (0,1)	3.013 (0.136)	3.327 ** (0.149)	4.140 (0.259)	4.552 (0.277)	1.840 (0.138)	2.160 (0.170)	0.324 (0.097)
	*0.0166*	*0.1878 ***	*−0.0691*	*0.0998*	*0.0324*	*−0.0995*	*0.1137*
**Sexual Orientation**							
Hetero sexuals (0,1)	3.132 * (0.075)	3.651 (0.081)	3.895 (0.145)	3.651 (0.081)	1.913 (0.084)	1.969 (0.092)	0.377 (0.0534)
Sexual Minori ties (0,1)	2.655 * (0.181)	3.609 (0.196)	4.375 (0.380)	3.609 (0.196)	1.958 (0.221)	1.958 (0.244)	0.600 (0.218)
	*0.1604 **	*0.0312*	*−0.0910*	*0.0312*	*−0.0182*	*0.0097*	*−0.0566*
		3.646 (1.033)	3.954 (1.900)	4.829 (2.022)	1.919 (1.063)	1.968 (1.165)	0.406 (0.830)
Total Sample	3.073 (1.070)						

^1^ For averages, standard errors are in parentheses; Spearman ps in italics (*p*-values not indicated are not significant); * *p* < 0.05; ** *p* < 0.01.

## Data Availability

Data is available by request from the corresponding author.

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
