# Peer review of "“Lovesick”: Mental Health and Romantic Relationships among College Students"

_ijerph, 2022, doi:10.3390/ijerph20010641_

Round 1

Reviewer 1 Report

dear authors thank you very much for the submission.

the paper has a three major limitations that need to be addressed:

first, there is big difference between psychiatric symptoms or mental health and psychiatric disorders. the fact you asked about psychiatric disorder with self-report this makes the study questionable. 

second, in light of above point obtaining prevalence rate of approximately 60% of students (70% women and 50% men) has a psychiatric disorder (major illnesses) does not make sense and concerning. this is clearly not psychiatrists diagnosed. there is big difference between symptoms diagnosis. 

third, the analysis is artificial and providing significant p values with no clear importance. with 276 participants sample size is underpowered.

many additional methodological concerns e.g. response rate, sampling, selection crietria, reliability and need for alpha and Omega coeeficienta and cfa metrics need to be addressed.

Author Response

Please see the attachment. We thank Reviewer 1 for their helpful feedback and insight into the paper.

Reviewer 2 Report

This study is interesting and has the potentials to be accepted. However, I would like to propose the following extensive modifications, which can enhance the reliability and validity of the authors’ study:

1.       Research questions, that drive the paper, should be built in the introduction from an ongoing and pertinent bibliography (up to 2022-23) and these should be of global interest and not focused on a particular local problem. Identifying a research gap is the most important by indicating in-text some newer references that are significant to your particular field of research.

2.       The problem statement needs to be clear into Introduction. Does the relevant literature refer the potentials of using “conventional” instead of typical VR-supported activities to drive us there?

3.       Are there any literature reviews to give us a point of view about this investigation? In other words, the authors need to “combine” any key word that investigate with any relevant study that can be integrated in-text as “background”.

4.       Any information respecting participants’ background can be beneficial for readers to understand better the level of technological literacy can overcome this “novelty effect”.

5.       The added value of authors’ work is not clear in terms of proper and current research (up to 2022-23), because several references are not up-to-date.

6.       Authors should answer your research question in the conclusions and discussion. Please provide a reasonable need to read your work’s results than previous ones or simply answer what we learned compared with current, significant research (up to 2022 should be your work’s “significance”).

7.       How general are your results and how do you believe that such findings have to be of global interest? Please relate these with your limitations and Discussion that is not exist. Why?

8.       Practical and educational implications are not provided.

Author Response

Please see the attachment. We thank Reviewer 2 for their helpful feedback and insight into the paper.

Round 2

Reviewer 1 Report

there remain major concerns about the methods used. 

dsm criteria changed over versions and there is a clear self selection bias in the sample. 

the results do not make sense and an alpha of 0.6 is considered as unacceptable or poor at best. 

the conclusions dont correspond to the data. 

Reviewer 2 Report

The authors have satisfactorily responded to all my questions and made the necessary changes to the manuscript.
